# Are the Modern Diets for the Treatment of Obesity Better than the Classical Ones?

Chrysi C. Koliaki * and Nicholas L. Katsilambros

First Propaedeutic Department of Internal Medicine and Diabetes Center, Laiko General Hospital, Medical Faculty, National and Kapodistrian University of Athens, 15772 Athens, Greece
* Correspondence: ckoliaki@yahoo.com

**Abstract:** Conventional hypocaloric diets, providing continuous energy restriction, are considered to be the cornerstone of dietary management of obesity. Although energy-restricted diets are overall safe, healthy, and modestly effective, their long-term adherence is difficult to accomplish. Intermittent fasting and ketogenic diets have emerged as attractive alternative dietary options for weight loss and improvement in cardiometabolic risk. Intermittent fasting is a unique dietary pattern characterized by periods of eating alternated with periods of fasting. Ketogenic diets are very low in carbohydrate, modest in protein, and high in fat. Several systematic reviews and meta-analyses of randomized controlled trials (RCTs) have reported beneficial but short-lived effects of intermittent fasting and ketogenic diets on various obesity-related health outcomes. Although for both diets, the current evidence is promising and steadily evolving, whether they are better than traditional calorie-restricted diets, whether they can safely lead to sustained weight loss and overall health benefits, and their effects on body composition, weight loss maintenance, energy intake and expenditure, diet quality, and cardiometabolic risk factors are still not unequivocally proven. The aim of the present review is to summarize the current state of evidence regarding the effects of these two popular modern diets, namely intermittent fasting and ketogenic diets. We describe the rationale and characteristics of different dietary protocols, we analyze the major mechanisms explaining their weight loss and cardiometabolic effects, and we provide a concise update on their effects on body weight and cardiometabolic risk factors, focusing on meta-analyses of RCTs. We also discuss knowledge gaps in the field of these diets, and we indicate directions for future research.

**Keywords:** intermittent fasting; alternate-day fasting; time-restricted eating; ketogenic diets; weight loss; cardiometabolic risk factors





## 1. Introduction

Long-term weight control with safe and effective dietary strategies is critical to reducing the exponentially increasing prevalence of excess body weight worldwide and reduce obesity-associated health risks [1]. Obesity and overweight affect together over a third of the world's population today, and if current trends continue, an estimated 38% of the world's adult population will be overweight and another 20% will be obese by the year 2030 [2]. It is widely accepted that the first-line treatment of obesity is dietary management combined with behavior modification. The optimal diet to treat obesity remains a challenge, but as a general principle, it should be safe, effective, nutritionally balanced, and should facilitate long-term compliance and maintenance of weight loss [3]. Healthy dietary patterns are generally restricted in saturated fat, refined carbohydrates, and highly processed foods, and include instead fruits, vegetables, whole-grain foods, and low-fat dairy products, without necessarily counting calories on a daily basis [3].

Conventional hypocaloric diets, providing continuous (daily) energy restriction, are considered to be the cornerstone of dietary management of obesity. In the past, common versions of hypocaloric diets were low-fat diets with a macronutrient composition of 30%

fat, 50% carbohydrate and 20% protein [3]. It is important that these diets are individualized based on the weight loss course of each subject, and individual food preferences should be considered, since these diets are usually followed for long periods of time. Although energy-restricted diets are overall safe, healthy and modestly effective for short-term weight loss [4], the individual response to hypocaloric diets is heterogeneous and long-term adherence is difficult to accomplish [5].

Intermittent fasting diets have gained popularity over the past decade due to their remarkable simplicity, since they do not require subjects to track their total number of calories and strictly monitor their food intake on a daily basis. They are also relatively low-cost interventions, as opposed to traditional calorie-restricted dieting which is resource-intensive and difficult to sustain over time. Intermittent fasting is a unique dietary strategy characterized by periods of eating alternated with periods of fasting [6]. Several meta-analyses of randomized controlled clinical trials (RCTs), aiming to synthesize the available evidence in humans and assess its strength and quality, have reported a number of beneficial effects of intermittent fasting on various obesity-related health outcomes, including weight loss and cardiometabolic markers [7–12].

Ketogenic diets have been applied since 1920s as a therapy for medication-refractory epilepsy in children. From the 1960s onwards, they have become widely popular as one of the most common dietary approaches for weight management. Ketogenic diets are generally very low in carbohydrate, modest in protein, and high in fat content. They aim to induce ketosis, providing an alternative source of energy for cells that cannot directly metabolize fatty acids [13]. A number of systematic reviews and meta-analyses have demonstrated that ketogenic diets are associated with beneficial weight loss and metabolic effects in patients with obesity and type 2 diabetes mellitus (T2DM) in the short term [14–17].

Although both intermittent fasting and ketogenic diets have emerged as promising strategies for weight reduction and cardiometabolic risk improvement, whether they are significantly better than traditional calorie-restricted diets, whether they can safely lead to sustained long-term weight loss and overall health benefits, and their effects on body composition (mainly lean body mass preservation), weight loss maintenance, energy intake and expenditure, diet quality and cardiometabolic risk, are still not completely clear. To date, there has been extensive literature trying to address some of these aspects, but nevertheless, some degree of uncertainty still remains, especially with regard to long-term safety issues.

The aim of the present narrative review is to summarize the current state of evidence regarding the effects of two of the most popular modern diets, namely intermittent fasting and ketogenic diets. Intermittent fasting and ketogenic diets were particularly selected on the basis of the existing evidence that (a) intermittent fasting has been associated with unique metabolic benefits which are not so prominent in other types of diets [6], and that (b) ketogenic diets, although useful for specific indications in the short term, can pose considerable health risks in the long term [13]. In this article, we describe the rationale and characteristics of different dietary protocols, we analyze the major mechanisms explaining how they result in weight loss and positive cardiometabolic effects, and we provide a concise update on their effects on weight loss and cardiometabolic risk factors, focusing on meta-analyses of RCTs, which represent the highest quality of available scientific evidence. We also pinpoint knowledge gaps and unanswered questions in the field of these diets, and we indicate directions for future research. The major contribution of this review lies in the critical perspective it takes against modern diets, aiming to underline not only their positive but also their negative aspects and convey the message that enthusiasm should not outpace evidence.

## 2. Definition of Intermittent Fasting and Different Types

Intermittent fasting is an umbrella term used to denote various different dietary patterns. It is based on the simple principle of alternating periods of fasting (fast) with periods of unrestricted eating (feast). Fasting may range from zero-calorie consumption

(null eating) to significantly reduced energy intake (by 60–70%) [6]. Three different protocols of intermittent energy restriction have been mostly studied to date: alternate-day fasting (ADF), the 5:2 diet or else periodic fasting, and time-restricted eating (TRE) [6]. ADF or else every-other-day fasting (EODF) involves a day of fast alternating with a day of ad libitum food intake. On fast days, individuals can either consume only water and zero-calorie beverages which is termed zero-calorie or complete ADF [18,19], or alternatively, consume 25% of their total daily energy requirements (500–600 kcal per day), which is called modified ADF [20–22]. The 5:2 diet is a weekly time-restricted dietary pattern, which involves two fast days (consecutive or non-consecutive) and five feast days per week [23–25]. On the fast days, daily caloric intake is reduced, usually ranging between 500 and 1000 kcal per day. For ADF and the 5:2 diet, the timing of meals on the fast days is optional, but it is usually suggested that food intake should take place between 5.00 p.m. and 7.00 p.m. [26]. TRE represents a pattern of temporally restricted food intake without caloric restriction. It does not require reducing the overall caloric intake throughout the day, but requires compressing the window of energy consumption within a specified time frame within the day, with a large part of the food restriction taking place overnight during sleep. TRE involves restricting the eating window to a prespecified number of hours per day (usually 4 to 8 h), and fasting with water or zero-calorie beverages for the rest of the day [27–31]. In the majority of trials, the suggested eating window in TRE interventions is 8 h. Early TRE (eTRE) indicates restricting food intake in the morning window, namely have the last meal not beyond 3.00 p.m. Table 1 presents the major types of intermittent fasting studied in the literature, and describes their characteristics.

**Table 1.** The most widely studied dietary regimens of intermittent energy restriction.

| Dietary Pattern | Description |
| --- | --- |
| Zero-calorie ADF | One day complete fasting (only water and zero-calorie beverages) and the other day eating without restrictions (ad libitum) |
| Modified ADF | One day reduced caloric intake to 25% of total daily energy needs (no more than 500–600 kcal/day) and the other day eating without restrictions (125% of total daily energy needs) |
| 5:2 diet | 2 days per week reduced caloric intake (500–1000 kcal/day) and the other 5 days per week eating without restrictions (ad libitum) |
| TRE | Fasting for at least 12 h per day, food intake restricted only within prespecified time windows of 4, 6, 8, 10 or 12 h per day, and water and zero-calorie beverages on the remaining hours of the day. The most prevalent pattern is 16:8 (fast 16 h, eat 8 h) |
| Early TRE (eTRE) | Food intake restricted in the morning window (last meal of the day before 3.00 p.m.) |

ADF: alternate-day fasting; TRE: time-restricted eating.

## 3. Mechanisms Explaining the Beneficial Metabolic Effects of Intermittent Fasting

The beneficial effects of intermittent fasting on cardiometabolic health are mediated by mechanisms associated with circadian rhythms, glucose-to-ketone metabolic switch, mitochondrial function, adipose tissue function and gut microbiota alterations [32].

Energy restriction for more than 10 h triggers a metabolic switch from liver-derived glucose to adipose tissue-derived free fatty acids (FFAs) and ketones. Cells are able to adapt to this bioenergetic challenge (glucose depletion, oversupply of FFAs/ketones) by activating protective signaling pathways that up-regulate mitochondrial function and antioxidant defenses, while up-regulating autophagy to remove damaged molecules and recycle their components [32]. Periodic flipping of the metabolic switch not only provides the ketones that are important fuels for cells during fasting, but also elicits highly orchestrated systemic and cellular adaptive responses that remain active also in the fed state and confer chronic disease resistance [33,34]. Ketones are not only energy fuels but also potent signaling molecules with important effects on multiple cell functions [35]. Ketone bodies regulate the expression and activity of many proteins known to influence health and disease, including

peroxisome proliferator-activated receptor γ coactivator 1α (PGC-1α), fibroblast growth factor-21 (FGF-21) and sirtuins [36,37].

Based on a large body of evidence, mainly consisting of preclinical studies in animal models, intermittent fasting induces a variety of evolutionarily conserved cellular metabolic adaptations which are mainly summarized in the following [32,38,39]: increased hepatic and skeletal muscle glycogen breakdown, inhibited anabolic processes such as lipogenesis and protein synthesis, increased FFA oxidation, increased ketogenesis, improved postprandial lipid metabolism, improved glucose regulation, resistance to metabolic stress, enhanced mitochondrial biogenesis and uncoupling, reduced reactive oxygen species (ROS) production, suppressed inflammation, improved cell survival, reduced leptin and increased adiponectin secretion, and stimulated quality control mechanisms such as autophagy.

Another proposed mechanism which could possibly mediate the beneficial metabolic effects of intermittent fasting and warrants further investigation relates to beige adipose tissue development and modifications of gut microbiota. An interesting experimental study in mice has shown that ADF was able to selectively stimulate beige fat development within white adipose tissue (WAT) and ameliorate obesity, insulin resistance and hepatic steatosis to a profound extent [40]. In this sophisticated animal study, ADF could promote WAT beiging by inducing specific alterations in gut microbiota composition, leading to elevated levels of the fermentation products acetate and lactate which acted as beiging agents [40]. Of note, ADF did not activate brown adipose tissue (BAT) in this study, but only stimulated the beige transformation of WAT, and increased energy expenditure through non-shivering thermogenesis [40]. Given that total food intake was not affected, the observed dramatic weight loss was attributed to the increase in energy expenditure due to beiging.

It has been suggested that TRE is beneficial for metabolism mainly due to effects on circadian biology [41]. The metabolic effects of TRE appear to depend on the time of day of the eating window in humans. Restricting food intake to the middle of the day (mid-day TRE) has shown the potential to reduce weight, body fat, fasting glucose and insulin levels, insulin resistance, hyperlipidemia and inflammation [42,43]. On the other hand, restricting food intake to the late afternoon after 4.00 p.m. (late TRE) has been shown to produce either null results or even worsen postprandial glucose and lipid metabolism, pancreatic β-cell responsiveness and blood pressure [28,44,45]. The circadian system, or else the internal biological clock, may explain why the effects of TRE appear to largely depend on meal timing. Circadian rhythms orchestrate metabolism by temporally separating opposing metabolic processes and anticipating recurring feeding–fasting cycles to increase metabolic efficiency [46]. In humans, insulin sensitivity, β-cell responsiveness and meal-induced thermogenesis are all higher in the morning than in the afternoon/evening, suggesting that human metabolism is optimized for food intake in the morning [46–48]. There is also accumulating evidence that circadian misalignment induced by an abnormal timing of light exposure, sleep or food intake may adversely affect metabolic health in humans [49,50]. This complex interconnection between circadian rhythmicity and systemic metabolism, underscores the importance of chronobiology of nutrition for preventing and treating obesity and metabolic diseases.

With regard to the effects of TRE on blood pressure, it has been postulated that possible underlying mechanisms involve the increased parasympathetic nerve activity due to elevated brain-derived neurotrophic factor (BDNF), the increased norepinephrine excretion by the kidneys, and the increased sensitivity to natriuretic peptides and insulin [51].

As far as effects on whole-body insulin sensitivity are concerned, it has been hypothesized that the prolonged fast of TRE regimens is expected to lead to a pronounced depletion of hepatic glycogen stores overnight and might improve insulin sensitivity due to an increased need to replenish nutrient storage. However, this pathophysiological concept has not been confirmed by clinical data. In a 3-week, randomized, cross-over study in 14 obese patients with T2DM, the TRE intervention (10 h eating window) had no significant effect on hepatic glycogen content, hepatic and peripheral insulin sensitivity, skeletal muscle mitochondrial function and whole-body energy metabolism, but was associated

with an improvement in fasting and 24 h glucose levels assessed by continuous glucose monitoring [52]. Furthermore, daily energy expenditure was not affected, glucose oxidation decreased, and insulin-induced non-oxidative glucose disposal was increased [52].

## 4. The Current State of Evidence on the Effects of Intermittent Fasting

A recent narrative review assessed the effects of various intermittent-fasting protocols on body weight and cardiometabolic parameters in humans [26]. The authors critically reviewed the data from six RCTs on ADF [19,21,22,53–55], seven RCTs on the 5:2 diet [23–25,56–59] and nine RCTs on TRE with either a 4, 6 or 8 h eating window [27–30,43,44,60–62], and provided some useful take-home messages with regard to the effects of intermittent fasting on weight loss, body composition, energy intake, diet quality and cardiometabolic risk markers [26]. The major conclusions drawn in this comprehensive review are graphically illustrated in Figure 1 and they are summarized in the following [26]: (i) ADF and the 5:2 diet can lead to mild to moderate weight loss in the short-term in obese humans (4–8% vs. baseline over 8–12 weeks), while TRE has been associated with cardiometabolic improvements but does not result in clinically significant weight loss (only 3–4% over 8–12 weeks); (ii) the weight loss efficacy of ADF and the 5:2 diet seems to peak at 12 weeks and declines thereafter; (iii) intermittent fasting produces comparable weight loss with the conventional dietary approach of daily caloric restriction (DCR), and is thus not superior; (iv) the weight loss effects of intermittent fasting are not gender-specific (males and females experience similar weight losses) and they extend to several different populations, including those with obesity, insulin resistance, prediabetes, type 1 (T1DM) and T2DM to a similar extent, (v) both ADF and the 5:2 diet facilitate weight loss maintenance, since they prevent weight regain over 12–24 weeks of follow-up; (vi) intermittent fasting diets are not superior to traditional hypocaloric diets in terms of lean body mass (LBM) preservation and visceral fat mass (VFM) reduction, since both approaches of energy restriction (continuous and intermittent) are associated with the same degree of LBM loss (25%) and VFM reduction in the setting of equivalent weight loss; (vii) the overall energy intake is reduced by approximately 10–30% relative to baseline in subjects following intermittent-fasting diets, while energy expenditure remains unchanged (net energy deficit); (viii) macronutrient composition and diet quality are not affected by intermittent fasting compared to the baseline state, while effects on micronutrient intake (vitamins and minerals) have not been adequately studied; (ix) intermittent fasting has been associated with neutral or beneficial effects on numerous cardiometabolic risk factors including blood pressure (reductions by 3–13%), triglycerides (reductions by 16–36%), low-density lipoprotein (LDL) cholesterol levels (reductions by 10–22%), fasting insulin levels (reductions by 11–38%), and circulating markers of oxidative stress, while fasting glucose levels (in non-diabetic individuals) and circulating markers of subclinical inflammation such as C-reactive protein (CRP), tumor necrosis factor $\alpha$ (TNF-$\alpha$) and interleukin-6 (IL-6) are not appreciably altered, whereas high-density lipoprotein (HDL) cholesterol levels are slightly decreased; (x) insulin resistance and glycemic control (assessed by glycated hemoglobin HbA1c) can be improved with intermittent fasting, primarily in insulin-resistant patients with diabetes at baseline.

A number of systematic reviews and meta-analyses of RCTs have been performed with the aim to collectively analyze the existing data on the weight loss and cardiometabolic effects of intermittent-fasting diets in obese humans with and without diabetes. These data represent the highest quality of available scientific evidence in the field of intermittent fasting and are summarized in Table 2. In some meta-analyses, the focus of interest was the short-term efficacy of intermittent fasting as an alternative approach for weight management in obese adults [63], while in others, longer-term effects were analyzed using RCTs of at least 6 months duration [64]. The common conclusive statement in all these meta-analyses is that intermittent fasting is more effective than routine diet (RD) (lack of any intervention) for weight loss, but does not provide better weight loss or cardiometabolic outcomes compared to the conventional approach of DCR [65]. It should be thus recommended as an alternative weight loss strategy for those subjects

having difficulties in adhering to continuous energy-restricted diets due to restrictions in dietary choices.

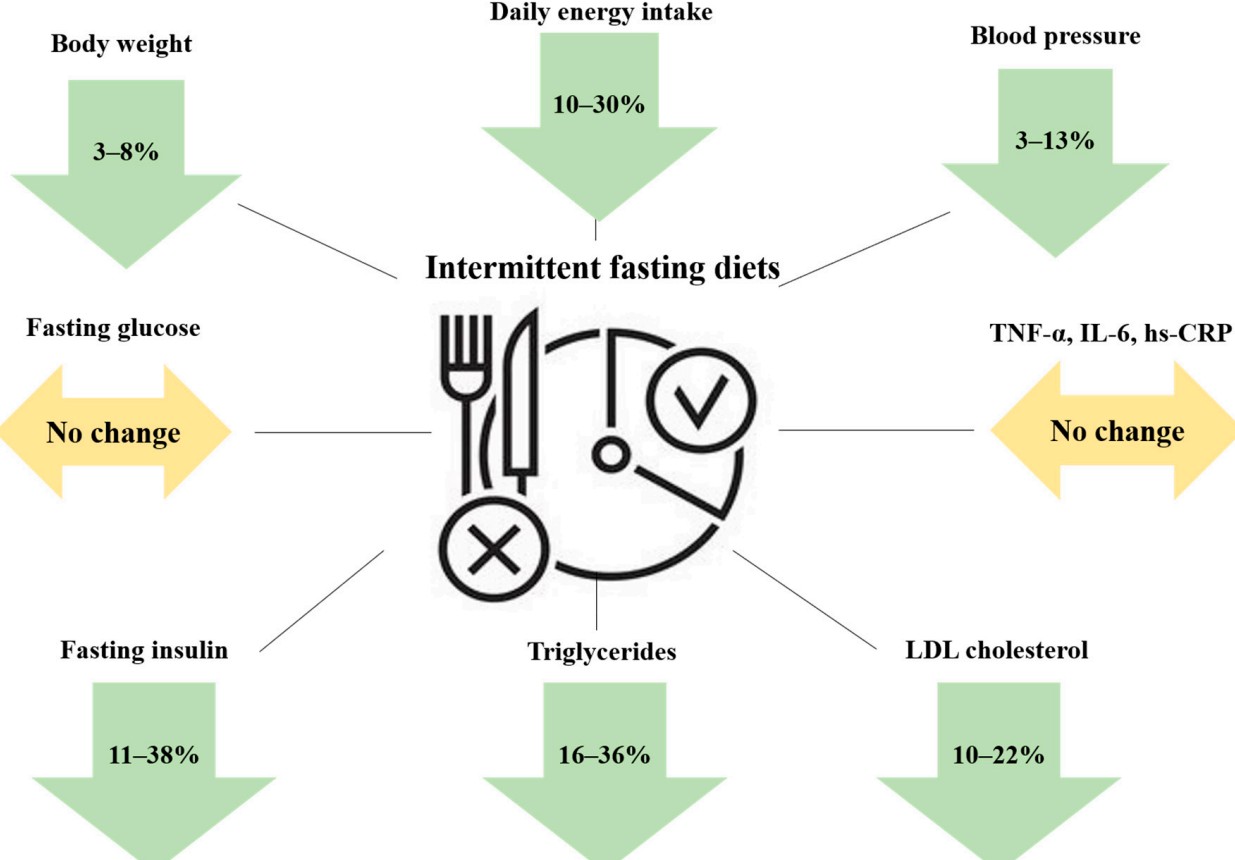

**Figure 1.** A graphical summary of the major effects of intermittent-fasting diets on body weight and cardiometabolic risk factors. Abbreviations: hs-CRP: high-sensitivity C-reactive protein; IL-6: interleukin-6; LDL: low-density lipoprotein; TNF-α: tumor necrosis factor α. The downward arrows indicate a decrease, while the horizontal arrows indicate lack of change in the respective parameters.

In one of these meta-analyses, a subgroup analysis revealed gender-specific effects of intermittent fasting compared to both RD and DCR [65]. More specifically, intermittent fasting was found to reduce more effectively fat mass in women, while the effects on body weight and triglyceride levels were more pronounced in men [65]. These data come in contrast with the previously discussed narrative review by Varady et al., which failed to demonstrate any gender-specific differences and suggested that both men and women display similar weight loss benefits with intermittent fasting [26]. An additional discrepancy between these two review papers is related to the potential of intermittent fasting to preserve LBM. The narrative review of Varady et al. suggested that intermittent fasting leads to similar LBM loss as DCR [26], while the systematic review of Gu et al. concluded that LBM is preserved or even increased with intermittent fasting, a finding, however, having only marginal statistical significance.

A common conclusion drawn by most meta-analyses published to date is the suboptimal quality of the available evidence as assessed by the GRADE system and several risk-of-bias assessment tools. Of note, an umbrella review of 11 meta-analyses of a total number of 130 RCTs assessing the most representative types of intermittent fasting (zero-calorie ADF, modified ADF, 5:2 diet and TRE) found that only six associations had moderate–high quality of evidence [66]. High-quality evidence was demonstrated only for modified ADF, which was associated with moderate weight reduction in healthy adults

as well as overweight/obese and patients with non-alcoholic fatty liver disease (NAFLD) compared to RD over a short-term period of 1–2 months [66].

Until recently, the literature lacked a well-designed RCT directly comparing a TRE intervention with DCR over a period of more than 12 weeks. A single such study recently published, conducted in a Chinese metabolically healthy obese population, addressed this gap and reported that a calorie-restricted TRE diet in the form of restricting food intake in an 8 h window between 8.00 a.m. and 4.00 p.m. (early TRE + caloric restriction) did not provide additional benefits over DCR alone (without meal timing restrictions) in terms of body weight reduction and improvement in cardiometabolic risk factors over a period of 12 months [67]. Although this study has been criticized with the arguments that it applied a mild TRE intervention deviating only by 2 h from the habitual 10 h eating window of Chinese individuals, that reduced caloric intake in the TRE group is likely to have blunted any effect of TRE independent of caloric restriction, and finally that the study population comprised healthy subjects leaving too little space for improvements [68], there is no doubt that it is an important study in the field of intermittent fasting. It adds another robust piece of evidence to the available literature which fails to substantiate any clinically meaningful differences (superiority) of intermittent fasting over the established approach of DCR in terms of weight reduction and improvement in metabolic risk [68].

More research is needed in order to conclude on the effects of intermittent fasting on markers of inflammation. Although the review of Varady et al. suggested that there is no impact of intermittent fasting on markers of subclinical inflammation [26], a recent RCT reported an improvement in markers of adipose tissue dysfunction [69]. In the latter study, 28 Norwegian obese adults (predominantly women) were randomized to either intermittent fasting (550–660 kcal per day on 3 non-consecutive days per week) or DCR for 12 weeks. A large number of circulating pro-inflammatory cytokines and chemokines, excluding IL-6 and TNF-$\alpha$, were decreased only in the intermittent-fasting group, which was paralleled by reduced insulin resistance and comparable weight loss relative to the DCR group [69]. According to a systematic review and meta-analysis focusing on the effects on inflammatory markers, the characteristics of the study population and the length of treatment duration play an important role in determining the magnitude of these effects, since CRP reduction was found to be more pronounced with intermittent fasting vs. DCR in overweight/obese subjects and those following the diets for more than 8 weeks [70].

Safety issues: Intermittent fasting is generally considered a safe dietary approach but may occasionally give rise to gastrointestinal, neurological, hormonal or metabolic adverse effects [26]. Headaches have been reported during the first two weeks of fasting due to dehydration [27,71], but can be corrected with sufficient fluid intake. Intermittent fasting can suppress metabolism and decrease resting metabolic rate during weight loss, to a similar extent as DCR [22,58]. It remains also unknown whether intermittent fasting is safe in subjects with diagnosed eating disorders. Furthermore, the effects of fasting on fertility remain unknown, as no trial has been conducted to date to address this issue.

Practical implementation and adherence: Intermittent fasting should not be prescribed to children or pregnant/lactating women, as no study to date has evaluated the safety of these diets in these populations. It is further not recommended in underweight or malnourished subjects and people with a history of eating disorders. Caution is warranted for elderly individuals, considering the unclear effects of fasting on aging-related sarcopenia. Overweight/obese healthy adults without comorbidities can safely undertake these diets without medical supervision.

During an intermittent-fasting intervention, patients should be encouraged to consume healthy and nutrient-dense foods such as fruits, vegetables and whole-grain products to enhance their fiber and micronutrient intake. It is also recommended to consume at least 50 g of lean protein on the fast day of ADF and the 5:2 diet to help control hunger [72–74] and prevent excessive LBM loss [75,76]. Clinicians should regularly assess the frequency of adverse effects during the first 3 months, monitor for nutrient deficiencies and assess the

continuing need for glucose-lowering, hypolipidemic and anti-hypertensive medications (potential need for dose adjustment or treatment de-escalation).

With regard to ease of compliance, feasibility and long-term sustainability, data from RCTs contradict the widely held expectation that intermittent-fasting diets are simpler to implement and thus easier to follow compared to continuous hypocaloric diets. In particular, it has been shown that modified ADF produces no superior adherence in the long term (significantly higher drop-out rates compared to DCR; 38 vs. 29%) [22]. It has been further shown that the adherence to the 5:2 diet, a diet considered to be more straightforward and less demanding than DCR and is possibly appropriate for people with high levels of stress and limited resources, may be initially high (74% at 6 weeks), but declines significantly thereafter (31% at 6 months and only 22% at 12 months) [77]. Intermittent fasting appears to be mostly successful in the initial phase of 1–6 months, but then participants experience often a plateau and additional weight loss cannot be achieved due to compensatory adaptations resisting further weight loss, which are also encountered in DCR, or decreased adherence to the assigned weight loss strategy [64,78,79].

A limited number of pilot studies applied novel technologies, especially smartphone applications, in order to identify subjects with abnormal meal timing patterns and also implement interventions aiming to modulate these erratic patterns of diurnal eating [42,80]. These studies found important long-term beneficial effects of technology-based TRE interventions on several health markers, and should thus pave the way for further such innovative studies in the future.

**Table 2.** A summary of the major systematic reviews and meta-analyses of RCTs in the field of intermittent fasting in overweight and obese humans.

| Author (Year) | Number of RCTs Included | Total Number of Participants Study Population Characteristics | Dietary Intervention vs. Comparator | Duration of Intervention | Major Outcomes |
|---|---|---|---|---|---|
| Headland et al. (2016) [64] | 6 | *n* = 981 Overweight/obese ± T2DM | IER vs. DCR | ≥6 months (weight loss and maintenance phase) | Similar effects on BW, serum lipids, FPG, FPI Similar drop-out rates |
| Cioffi et al. (2018) [7] | 11 | *n* = 630 Overweight/obese ± T2DM Predominantly healthy obese women | Several versions of IER (1–6 fast days per week) vs. DCR | 8–24 weeks | Similar effects on BW Slight ↓ FPI (of no clinical value) Heterogeneous adherence Often higher attrition rates vs. DCR |
| Harris et al. (2018) [63] | 5 | *n* = 376 Overweight/obese Females 79% | Weekly IER (≥7 days) vs. DCR | 14–48 weeks (mean 26) | Comparable weight loss (>5 kg) and health benefits over a 3–6-month period |
| Harris et al. (2018) [8] | 6 | *n* = 400 Overweight/obese Non-diabetic | IER (ADF, periodic fasting patterns with fasting 2–4 days per week) vs. RD or DCR | 3–12 months (mean 5.6) | Compared to RD: Better weight loss Compared to DCR: Similar effects on BW ↓ WC, ↓ FM ↓ FPI ↔ serum lipids and lipoproteins ↔ FPG |

**Table 2.** *Cont.*

| Author (Year) | Number of RCTs Included | Total Number of Participants Study Population Characteristics | Dietary Intervention vs. Comparator | Duration of Intervention | Major Outcomes |
|---|---|---|---|---|---|
| Roman et al. (2018) [10] | 9 | *n* = 782 Overweight/obese | IER (5:2 diet, modified ADF) vs. DCR | 1–12 months | ↓ LBM vs. DCR Similar effects on BW, FM, WC |
| Cho et al. (2019) [9] | 12 | *n* = 545 Normal-weight, overweight, obese Non-diabetic | IER (ADF, modified ADF, 5:2 diet, TRE) vs. RD or DCR | 4–24 weeks | ↓ BMI, FPG, HOMA-IR Trend for ↓ FM ↓ leptin, ↑ adiponectin Preserved LBM |
| Cui et al. (2020) [11] | 7 | *n* = 269 Normal-weight, overweight/obese adults | Modified ADF vs. RD | 1–12 months | ↓ BW, BMI ↓ FM, LBM ↓ TC, LDL, TG ↓ SBP, DBP ↓ total caloric intake ↔ HDL, FPG, HOMA-IR |
| Meng et al. (2020) [81] | 28 | *n* = 1528 Normal-weight, overweight/obese adults | IER (5:2 diet, modified ADF) vs. RD | 1–12 months | ↓ TC, LDL, TG ↔ HDL |
| Moon et al. (2020) [82] | 12 | *n* = 328 A broad range of participants: normal-weight healthy young adults, young active females, healthy active males, prediabetic men, overweight/obese, NAFLD | TRE vs. RD | 4 days–12 weeks | ↓ BW, ↓ FM Preserved LBM ↓ SBP ↓ FPG ↓ TG ↔ LDL, HDL |
| Park et al. (2020) [83] | 8 | *n* = 728 Normal-weight, overweight/obese adults | Modified ADF vs. RD, DCR or TRE | 1–8 months | For overweight subjects and <6 months duration: ↓ BW, BMI, FM ↓ TC For obese subjects >40 years old: ↓ WC |
| Pellegrini et al. (2020) [84] | 11 | *n* = 452 Healthy adults or with chronic diseases (not affecting outcomes) | TRE (Ramadan fasting) vs. RD or DCR | 4–8 weeks | ↓ BW ↓ LBM ↓ FPG |

**Table 2.** *Cont.*

| Author (Year) | Number of RCTs Included | Total Number of Participants Study Population Characteristics | Dietary Intervention vs. Comparator | Duration of Intervention | Major Outcomes |
|---|---|---|---|---|---|
| Wang et al. (2020) [70] | 18 | *n* = 920 Over-weight/obese, healthy subjects, patients with obstructive sleep apnea, rheumatoid arthritis | IER vs. DCR | NA | ↓ CRP ↔ IL-6, TNF-α More pronounced CRP reduction in overweight/obese and for interventions lasting ≥ 8 weeks |
| Welton et al. (2020) [85] | 18 | *n* = 1490 Overweight/obese ± T2DM | Various types of IER (ADF, 5:2 diet, TRE) vs. DCR | 8–52 weeks | Similar effects on BW and BMI Improved glycemic control in T2DM patients No serious adverse events reported Heterogeneity of data |
| He et al. (2021) [12] | 11 | *n* = 850 Overweight/obese adults | IER (5:2 diet, modified ADF) vs. DCR | 3–12 months | ↓ BW ↓ HOMA-IR Especially in the short term (over 2–3 months) |
| Yang et al. (2021) [86] | 46 | *n* = 2681 A broad range of participants: lean, overweight, obese, sedentary, active, young, elderly, T2DM, NAFLD | Different types of IER (ADF, modified ADF, 5:2 diet, TRE etc.) vs. RD or DCR | 7 days– 12 months | ↓ BW, WC, FM, BMI, SBP, DBP, FPG, FPI, HOMA-IR, TC, TG ↔ LDL, HDL, HbA1c |
| Pureza et al. (2021) [87] | 9 | *n* = 184 Overweight/obese | eTRE vs. RD and other variants of TRE and meal timing patterns | 1 day– 12 weeks | ↓ FPG, ↓ HOMA-IR Publication bias, low-quality evidence |
| Gu et al. (2022) [65] | 43 | *n* = 2483 Normal-weight, overweight, obese ± T2DM ± NAFLD Mostly non-diabetic (metabolically healthy) | IER (ADF, 5:2 diet, TRE, Ramadan fasting) vs. RD or DCR DCR included also Mediterranean diet and DASH | ≥4 weeks (median 12) | Compared to RD: ↓ BW, BMI, WC, FM ↓ FPI, HOMA-IR ↓ TC, TG ↑ LBM ($p$ = 0.05) ↔ FPG ↔ SBP, DBP ↔ LDL, HDL Compared to DCR: ↓ WC, no other differences |

ADF: alternate-day fasting; BMI: body mass index; BW: body weight; DASH: dietary approach to stop hypertension; DBP: diastolic blood pressure; DCR: daily caloric restriction, eTRE: early time-restricted eating; FM: fat mass; FPG: fasting plasma glucose; FPI: fasting plasma insulin; HbA1c: glycated hemoglobin A1c; HDL: high-density lipoprotein cholesterol; HOMA-IR: homeostasis model assessment for insulin resistance; IER: intermittent energy restriction; IL-6: interleukin-6; LDL: low-density lipoprotein cholesterol; NA: not available; NAFLD: non-alcoholic fatty liver disease; LBM: lean body mass; RCTs: randomized controlled clinical trials; RD: routine (habitual) diet; SBP: systolic blood pressure; T2DM: type 2 diabetes mellitus; TC: total cholesterol; TG: triglycerides; TNF-α: tumor necrosis factor-α; TRE: time-restricted eating; VLCD: very-low-calorie diet; WC: waist circumference. ↑ indicates increase, ↓ indicates decrease and ↔ indicates no change.

## 5. The Rationale of Ketogenic Diets and Different Variants

The ketogenic diet is unique and distinct from other low-carbohydrate diets such as the Paleo or Atkins diet in that subjects are encouraged to forget nearly all carbohydrates, avoid excess protein, and consume high levels of fat (which generally exceed 70% of total calories consumed), resulting in the production of ketones [88].

The classic ketogenic diet is a high-fat, very-low-carbohydrate diet, which restricts carbohydrate intake to 5–10% of total daily energy intake and replaces the remaining with high amounts of dietary fat (70–80%) and moderate amounts of protein (10–20%) [89]. Depending on the exact weight ratio of dietary fat to combined protein and carbohydrate, it can be further classified into 4:1 and 3:1, and should be always medically supervised. Beyond the classic ketogenic diet, there are also other variants of low-carbohydrate diets which allow more protein or carbohydrate [90], as summarized in Table 3. Ketogenic diets are different from the so-called low-carbohydrate diets. The latter refer to a carbohydrate intake below the recommended dietary allowance of 130 g/day [17], which is usually not sufficiently low to induce nutritional ketosis, whereas in ketogenic diets, carbohydrate intake is restricted to less than 50 g/day.

**Table 3.** The major types of ketogenic diets.

| Variant | Description (Macronutrient Composition) |
| --- | --- |
| Classic ketogenic diet 4:1 | Fat 90%, CHO 2–4%, protein 6–8% |
| Classic ketogenic diet 3:1 | Fat 85–90%, CHO 2–5%, protein 8–12% |
| Modified Atkins diet | Fat 60–65%, CHO 5–10%, protein 25–35% |
| Ketogenic diet | CHO intake <50 g/day |
| (as tested in scientific studies) | Fat 70–80%, CHO <10%, protein 10% |
| Low-carbohydrate diet | CHO intake <130 g/day<br>Fat variable, CHO 10–25%, protein variable |

CHO: carbohydrates.

The major mechanism explaining the weight loss effects of ketogenic diets is associated with reduced insulin levels, which redirects lipid metabolism from storage to oxidation, promotes the use of ketone bodies as alternative fuels, and induces a metabolically beneficial state of nutritional ketosis, which mimics metabolic starvation in the human body [91]. Other possible mechanisms implicated in the weight loss effects of ketogenic diets include: (i) appetite suppression mediated by the higher satiety effect of proteins [92,93], the modulation of appetite-regulating hormones [94], and the direct anorexigenic effect of circulating ketone bodies [95]; (ii) reduced respiratory quotient ratio and greater metabolic efficiency in the direction of fat utilization [96]; (iii) increased metabolic costs of gluconeogenesis and protein-induced thermogenesis [97,98]; and (iv) increased energy expenditure as shown in short-term studies using state-of-the-art technology such as doubly labelled water [99,100].

## 6. The Current State of Evidence on the Effects of Ketogenic Diets

Table 4 summarizes the major meta-analyses of RCTs in the field of ketogenic diets in humans. In a meta-analysis of 13 RCTs assessing the long-term effects of ketogenic diets (intervention duration longer than one year), it was found that ketogenic diets were associated with less than a kilogram (0.9 kg) of additional weight loss as compared to high-carbohydrate low-fat strategies [17]. Although this difference was found to be statistically significant, it is doubtful whether it is also clinically significant and translates into a meaningful clinical effect. In the same meta-analysis, no significant difference was found in glycemic control (HbA1c) between ketogenic and low-fat diets in patients with T2DM, and there was also no difference in weight loss at 2 years [17].

Whether ketogenic diets are better than low-fat diets in terms of body composition changes promoting preferential fat mass loss remains questionable. In a metabolic ward study in 17 overweight or obese men, weight loss accelerated but fat mass loss slowed down for 2 weeks after switching from the baseline to the ketogenic diet [99]. A subsequent

metabolic ward study by the same research group tested the effects of an animal-based ketogenic diet compared to a plant-based, carbohydrate-rich, low-fat diet in 20 overweight young adults [100]. Participants were randomized to each diet, which they consumed for 2 weeks before crossing over to the other diet. Total energy intake was lower by nearly 700 kcal/day in the low-fat diet group. The reported hunger and dietary satisfaction scores were similar between groups. Both diets induced weight loss. However, most of the weight lost with the ketogenic diet came from LBM, whereas the low-fat diet resulted in significant losses of body fat after both the first and the second week [100]. The study concluded that low-fat, plant-based diets may control appetite and satiety even better than ketogenic diets contrary to what is commonly believed, and furthermore, that the rapid initial weight loss observed with ketogenic diets is predominantly due to loss of LBM comprising water, glycogen and muscle proteins [100]. These data fuel even more the skepticism regarding the superiority of ketogenic diets over the traditional low-fat hypocaloric dietary approaches.

Subgroup analyses have revealed more pronounced effects of ketogenic diets in overweight/obese and diabetic participants. A meta-analysis of 14 RCTs, comparing ketogenic diets with low-fat diets in subjects with and without T2DM, produced in summary the following findings: (i) ketogenic diets followed for 3–12 months are more effective than low-fat diets for improving glycemic control and insulin sensitivity in patients with T2DM, but their glycemic effects are comparable with those of low-fat diets in non-diabetic individuals; (ii) ketogenic diets followed for 1–12 months are more effective than low-fat diets for weight reduction in overweight and obese subjects, whether they have T2DM or not; (iii) ketogenic diets followed for 4 days up to 2 years have a more beneficial effect on lipid profile compared to low-fat diets only in diabetic patients, by reducing triglycerides and increasing HDL; but (iv) the lipidemic effect of ketogenic diets is adverse in non-diabetic subjects, by increasing total and LDL cholesterol [14]. The effects of carbohydrate-restricted diets on glycemic control in patients with T2DM appear to be most significant within the first 3–6 months and typically wane thereafter [15,101–103]. These effects are mainly attributable to weight loss [101,102]. Of note, there is too little evidence to support that ketogenic diets improve glucose intolerance independently of the concurrent weight loss. This is not the case with other dietary approaches such as low-fat diets, in which glycemic control is improved despite the consumption of healthy carbohydrate-rich foods, such as legumes, fruits and whole-grain products, even in the absence of weight loss.

In adults with T1DM, both beneficial and adverse outcomes have been reported. In a small study of 11 adults with T1DM, a ketogenic diet improved glycemic control, but triggered more frequent hypoglycemic episodes and promoted dyslipidemia [104]. In pathophysiological terms, sustained ketosis should be strongly discouraged in patients with T1DM, since it has been associated with oxidative stress, inflammation and insulin resistance [105]. In patients with T2DM participating in clinical trials with ketogenic diets, a common finding is that glucose-lowering medications are frequently reduced or even eliminated [106–113]. With regard to long-term glycemic effects, one open-label, non-randomized, controlled study of the ketogenic diet in patients with T2DM showed a reduction in HbA1c by 1.3% at one year and 0.9% at 2 years, but it should be noted that this group received intensive technological and behavioral support (digitally monitored continuous care intervention), which was not provided to the control group, and this is not always feasible in everyday clinical practice [112,114].

Safety issues and adherence: The potential adverse effects of ketogenic diets range from the relatively benign but unpleasant "keto flu", which is an induction period of fatigue, weakness and gastrointestinal disturbances, to the less common but more dangerous occurrence of cardiac arrhythmias due to selenium deficiency [88]. Other adverse effects may include nephrolithiasis, constipation, muscle cramps, headaches, bone fractures, pancreatitis, and multiple vitamin and mineral deficiencies [88]. In the absence of multivitamin supplementation, individuals on ketogenic diets are at risk of frank nutritional deficiencies [115]. Even when consuming nutrient-dense foods, the classic ketogenic diet 4:1 can

lead to multiple micronutrient shortfalls, often lacking in vitamin K, linolenic acid and water-soluble vitamins [116].

The most important risk, related to ketogenic diets, is the exclusion of fiber and unrefined carbohydrates from the diet, considering that fruits, legumes and whole grains have indispensable effects for human health. Extreme carbohydrate restrictions may profoundly affect diet quality, eliminating fiber sources and increasing the consumption of animal products being rich in cholesterol and saturated fat. In this context, evidence suggests that LDL cholesterol and apo-B-containing lipoprotein levels may fail to improve, or even significantly increase, with a ketogenic diet despite weight loss in healthy adults [117]. In a randomized, controlled, parallel-designed study in 30 young lean adults, a severely carbohydrate-restricted diet with less than 20 g carbohydrate per day for 3 weeks was associated with a significant increase in LDL cholesterol levels by 44% compared to the control habitual diet, and there was a greatly variable individual response [117]. Interestingly, carbohydrate-restricted diets can be linked to either increased or decreased mortality, depending on the quality of carbohydrates consumed and whether they rely more on animal proteins and saturated fat or plant-derived proteins and unsaturated fat, respectively [118]. These diets may also increase the risk of chronic diseases, since dietary components typically increased in ketogenic diets (red processed meat, saturated fat) are linked to an elevated risk of chronic kidney disease, cardiovascular disease, cancer, diabetes, and Alzheimer's disease [13].

**Table 4.** A summary of the major meta-analyses of RCTs in the field of ketogenic diets in humans.

| Author (Year) | Number of RCTs Included | Total Number of Participants Study Population Characteristics | Dietary Intervention vs. Comparator | Duration of Intervention | Major Outcomes |
|---|---|---|---|---|---|
| Nordmann et al. (2006) [119] | 5 | *n* = 447 Overweight/obese | Ketogenic diet vs. LFDs | ≥6 months | At 6 months with ketogenic diets: ↓ BW ↓ TG, ↑ HDL, ↓ TC, ↑ LDL At 12 months: ↔ BW |
| Hession et al. (2009) [16] | 13 | *n* = 1222 Overweight/obese | Ketogenic diet vs. LFDs | 6–36 months | ↓ BW, TG, SBP ↑ HDL ↓ Attrition rates vs. LFDs |
| Bueno et al. (2013) [17] | 13 | *n* = 1577 Overweight/obese Healthy, T2DM, CVD risk factors | VLCKD (<50 g CHO/day) vs. LFDs | 12–24 months | ↓ BW (0.9 kg) ↓ DBP ↓ TG, ↑ HDL, ↑ LDL |
| Meng et al. (2017) [120] | 9 | *n* = 734 T2DM | LCDs (<130 g CHO/day) vs. HCDs | 3–24 months | ↓ BW (1.18 kg) ↓ TG, ↑ HDL ↔ TC, LDL No weight loss effects in long-term RCTs (>12 months) |
| Sainsbury et al. (2018) [103] | 25 | *n* = 2412 T2DM | LCDs (≤45% CHO/day) vs. HCDs (>45% CHO/day) | 3–24 months | At 3 and 6 months: ↓ HbA1c (only in moderately CHO-restricted diets with CHO 26–45%) At 12 and 24 months: ↔ HbA1c |

**Table 4.** *Cont.*

| Author (Year) | Number of RCTs Included | Total Number of Participants Study Population Characteristics | Dietary Intervention vs. Comparator | Duration of Intervention | Major Outcomes |
|---|---|---|---|---|---|
| Chawla et al. (2020) [121] | 38 | *n* = 6499 Normal-weight, overweight, obese adults | LCDs (<40% CHO, >50% ketogenic) vs. LFDs | 1–24 months | Small ↓ BW (1.3 kg) with LCDs (6–12 months) ↑ Variability between studies and individuals |
| Choi et al. (2020) [14] | 14 | *n* = 734 Normal-weight, overweight and obese ± T2DM | Ketogenic diet vs. LFDs | 4 days–2 years | For T2DM patients: ↓ HbA1c, HOMA-IR (3–12 months) ↓ BW (1–12 months) ↓ TG, ↑ HDL (4 days–2 years) For non-diabetic subjects: ↔ HbA1c, HOMA-IR (3–12 months) ↓ BW (1–12 months) ↑ TC, LDL (4 days-2 years) |
| Goldenberg et al. (2021) [15] | 23 | *n* = 1357 T2DM | LCDs (<26% CHO/day) and VLCKDs (<10% CHO/day) vs. HCDs | 3–12 months | At 6 months with LCDs: ↑ T2DM remission rates ↓ BW, TG, HOMA-IR At 12 months with LCDs: -Sparse data on T2DM remission -Diminished effects on BW, TG, HOMA-IRVLCKDs less effective than LCDs for weight loss at 6 months ↓ adherence with VLCKDs vs. LCDs at 6 and 12 months |

BW: body weight; CHO: carbohydrates; CVD: cardiovascular disease; DBP: diastolic blood pressure; HbA1c: glycated hemoglobin A1c; HCDs: high-carbohydrate diets; HDL: high-density lipoprotein cholesterol; HOMA-IR: homeostasis model assessment for insulin resistance; LCDs: low-carbohydrate diets; LDL: low-density lipoprotein cholesterol; LFDs: low-fat diets; RCTs: randomized controlled clinical trials; SBP: systolic blood pressure; T2DM: type 2 diabetes mellitus; TC: total cholesterol; TG: triglycerides; VLCKDs: very-low-carbohydrate ketogenic diets.

Urinary ketone levels are often used as an indicator of dietary adherence [35]. Ketogenic diets may have poor long-term tolerability and sustainability for many individuals [101,102].

Taking all evidence into consideration, it can be concluded that the risks of ketogenic diets may actually outweigh the potential benefits, since ketogenic diets confer only temporary improvements, they have unfavorable effects on diet quality, and there are currently inadequate data confirming their long-term safety and sustainability [13].

## 7. Unanswered Questions and Directions for Future Research

In the field of intermittent fasting, there are several open questions which need to be addressed with high-quality research in the future. Some of these questions include the following [6,122]:

- Does TRE in the morning, afternoon or evening window have differential metabolic effects?
- Are there weight-loss-independent effects of intermittent fasting on cardiometabolic risk factors?

- Is there a specific intermittent-fasting protocol which is associated with better weight loss efficacy and long-term adherence?
- How do intermittent-fasting diets compare with healthy dietary patterns such as the Mediterranean diet in terms of weight loss and cardiometabolic risk improvement?

Studies aiming to dissociate the indirect effects of weight loss from the direct effects of fasting per se on cardiometabolic parameters are definitely warranted. In this context, there is a need for supervised controlled feeding trials investigating the direct metabolic effects of intermittent fasting independently of weight loss. One such proof-of-concept 5-week, randomized, cross-over study in eight overweight prediabetic men has shown that intermittent fasting in the form of an isocaloric early TRE intervention (6 h eating window from 8.00 a.m. to 2.00 p.m.) can improve a variety of cardiometabolic health markers including insulin sensitivity, postprandial insulinemia, pancreatic β-cell function, blood pressure and oxidative stress, even in the absence of significant weight loss [60]. It is difficult to draw meaningful conclusions regarding the superiority of one regimen of intermittent fasting over the other, given that there are no direct head-to-head trials comparing different types of intermittent fasting. The literature also lacks a direct comparative trial between intermittent-fasting diets and well-established healthy dietary patterns such as the Mediterranean or the DASH (dietary approach to stop hypertension) diet. It is important that such studies are designed and performed in the future. Common myths related to intermittent-fasting diets are that they increase energy expenditure (there is no evidence that they affect energy expenditure or adaptive thermogenesis), that they reduce hunger to a greater extent than traditional DCR (there are conflicting results in humans), and that they lead to compensatory overeating on the feast days (there is no evidence that they trigger eating disorders) [122].

Furthermore, more trials are needed to investigate the association of intermittent fasting with: (i) a broader range of populations and conditions such as adolescents, elderly people, cancer patients, and those with metabolic derangements beyond obesity such as polycystic ovary syndrome or thyroid disease, (ii) clinically relevant hard endpoints such as liver outcomes, malignancies, cardiovascular events, diabetes remission and prevention of diabetes in the setting of prediabetes, (iii) gut microbiome alterations and the impact on health outcomes, and (iv) short- and long-term safety outcomes such as adverse events, eating disorders, sleep quality, thyroid hormones and fertility. Data from adequately powered well-designed long-term RCTs are needed to elucidate the safety and efficacy of modern diets such as intermittent fasting and ketogenic diets in distinct population groups.

## 8. Concluding Remarks

Intermittent fasting and ketogenic diets can produce clinically significant weight loss in overweight and obese humans with and without diabetes, but not without limitations or question marks. Intermittent fasting produces comparable weight loss with the conventional dietary approach of DCR, but is not superior. The weight loss efficacy of intermittent fasting seems to peak at 12 weeks but declines thereafter. Intermittent fasting has been associated with beneficial effects on a variety of cardiometabolic risk factors in some studies, but not unequivocally. Insulin resistance and glycemic control can be improved with intermittent fasting, but primarily in diabetic patients. Ketogenic diets on the other hand can reduce body weight, but not more effectively than other dietary approaches in the long-term or when matched for energy intake. Ketogenic diets can also improve glycemic control in patients with T2DM, but their efficacy wanes after the first 3–6 months. Extreme carbohydrate restriction in ketogenic diets may profoundly affect diet quality, eliminating fiber sources and increasing the consumption of animal products which are rich in cholesterol and saturated fat. Carbohydrate-restricted diets can be linked to either increased or decreased mortality risk, depending on the quality of carbohydrates consumed. Taken altogether, the risks of ketogenic diets seem to outweigh the benefits, since ketogenic diets confer only temporary weight loss and metabolic improvements, have an unfavorable impact on diet quality, and there are also inadequate data confirming their

long-term safety and sustainability. For both modern diets discussed in this review, the current evidence is promising and steadily evolving, but data on long-term safety, efficacy, adherence, and superiority over the traditional approach of daily energy restriction are suboptimal, highlighting the need for more well-designed long-term RCTs testing these diets, and most importantly comparing them directly with each other, in several population groups. It is-well understood that this type of long-duration study is very difficult to perform in humans.

**Author Contributions:** C.C.K. performed the literature search and drafted the manuscript; N.L.K. coordinated writing, edited the manuscript and provided critical input at all stages of manuscript preparation. All authors have read and agreed to the published version of the manuscript.

**Funding:** This research received no external funding.

**Conflicts of Interest:** The authors declare no conflict of interest.

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
