# Peer review of "Are the Modern Diets for the Treatment of Obesity Better than the Classical Ones?"

_endocrines, doi:10.3390/endocrines3040052_

Round 1

Reviewer 1 Report

Overall the manuscript is well-written, and the topics are presented in a clear, understandable fashion.

Comments

11.   Line 49-50. It is stated that hypocaloric diets are usually low-fat. That was once accurate but not in 2022. A more recent reference supporting this point is needed. Or, revise to state that low-fat diets were very common.

22.     Line 79. The safety, efficacy, and long-term risks of the diets is certainly not clear. However, there is extensive literature on what is known.  Clarification is needed on this point

33.      There are other popular diets. Why focus on Time-restricted feeding and the Keto diet?

44.      Starting on line 496, the discussion is solely focused on intermittent fasting. Either the section title should be revised or discussion of the Keto diet added.

55.      In the analysis of future research, there is no mention of the appropriateness of the comparator diet.  It would be helpful if intermittent fasting or keto diets were compared to known healthy diets in which weight loss is also observed. (e.g. Med diet or DASH diet)

66.      With so many reviews published on diets and weight loss, it is important in the current manuscript to clarify why this manuscript is needed.

Author Response

Comments of Reviewer 1

1. Line 49-50. It is stated that hypocaloric diets are usually low-fat. That was once accurate but not in 2022. A more recent reference supporting this point is needed. Or, revise to state that low-fat diets were very common.

Authors' reply: We revised this sentence according to the reviewer’s suggestion, stating that hypocaloric diets were commonly represented by low-fat diets in the past (lines 49-51 in the revised manuscript, revision with red color).

2. Line 79. The safety, efficacy, and long-term risks of the diets is certainly not clear. However, there is extensive literature on what is known. Clarification is needed on this point. 

Authors' reply: We provide the clarification suggested in lines 80-83 of the revised manuscript (revision with red color). What is currently known regarding safety, efficacy and long-term risks of the reviewed diets is detailed and discussed analytically in the subsequent sections of the manuscript.

3. There are other popular diets. Why focus on Time-restricted feeding and the Keto diet?

Authors' reply: Thank you for this comment. Intermittent fasting and ketogenic diets were particularly selected as the focus of interest of this review among several other popular diets on the basis of the existing evidence that a) intermittent fasting has been associated with unique metabolic benefits which are not so prominent in other types of diets, and that (b) ketogenic diets, although useful for specific indications in the short term, can pose significant health risks in the long term. We provide this rationale and cite the respective references in the last paragraph of our Introduction, which describes the aim of the review (lines 86-90 of the revised manuscript, revision with red color).

4. Starting on line 496, the discussion is solely focused on intermittent fasting. Either the section title should be revised or discussion of the Keto diet added.

Authors' reply: Indeed, there are far more open questions in the field of intermittent fasting compared to ketogenic diets. In response to the reviewer’s comment, we revised the section title into “Unanswered questions and directions for future research” (page 16, line 521 of the revised manuscript, revision with red color). We deleted the first short paragraph of the original version of the manuscript which was about ketogenic diets, and we start directly with intermittent fasting and the related open questions (line 522 of the revised manuscript).

5. In the analysis of future research, there is no mention of the appropriateness of the comparator diet.  It would be helpful if intermittent fasting or keto diets were compared to known healthy diets in which weight loss is also observed. (e.g. Med diet or DASH diet)

Authors' reply: Thank you for this important comment. We absolutely agree with the reviewer that the appropriateness of the comparator diet is a significant aspect in the interpretation of the study results. In the present review, the vast majority of data presented comes from RCTs comparing intermittent fasting or ketogenic diets with the gold standard weight loss approach, which is hypocaloric diets with daily (continuous) energy restriction. We also agree that it would be meaningful to design and conduct RCTs comparing intermittent fasting or ketogenic diets with well-established healthy dietary patterns such as the Mediterranean or DASH diet. However, it should be noted that although these health-promoting diets have been associated with multiple cardiometabolic beneficial effects, their isolated effect on weight loss has been found to be relatively small (Lotfi K et al. Adv Nutr 2022;13:152-166). We refer to the need for future studies comparing intermittent fasting or ketogenic diets with healthy diets such as the Mediterranean or DASH diet in lines 531-532 and 544-547 of the revised manuscript (revision with red color).  

6. With so many reviews published on diets and weight loss, it is important in the current manuscript to clarify why this manuscript is needed.

According to the reviewer’s suggestion, we clarify what is the special aim and contribution of this review in our Introduction, lines 96-99 of the revised manuscript (revision with red color). The major contribution of this review lies in the critical perspective it takes against modern diets, aiming to underline (in a balanced way) not only their positive but also their negative aspects and ultimately convey the message that enthusiasm should not outpace the currently available evidence in this field.  

Reviewer 2 Report

-Having graphical information of weight/glycemic indices along with statistical induces would add more value and credibility to the study which is intended well. 

Author Response

Comments of Reviewer 2

Having graphical information of weight/glycemic indices along with statistical indices would add more value and credibility to the study which is intended well.

Authors' reply: We thank the reviewer for the helpful suggestion. We added Figure 1 in the revised version of our manuscript, which provides a graphical summary of the major effects of intermittent fasting diets on body weight and cardiometabolic risk factors. Figure 1 is cited on page 5, line 212, and can be found in detail on page 8 (legend on lines 354-355).